# Mapping the Cognitive Biases Related to Vaccination: A Scoping Review of the Literature

**DOI:** 10.3390/vaccines11121837

**Published:** 2023-12-11

**Authors:** Amar Raj, Awnish Kumar Singh, Abram L. Wagner, Matthew L. Boulton

**Affiliations:** 1Tata Memorial Hospital, Mumbai 400012, India; dramarraj@gmail.com; 2Former, National Technical Advisory Group on Immunization (NTAGI) Secretariat, Ministry of Health and Family Welfare, New Delhi 110011, India; 3School of Public Health, University of Michigan, Ann Arbor, MI 48109, USA; awag@umich.edu (A.L.W.); mboulton@umich.edu (M.L.B.)

**Keywords:** cognitive bias, vaccination, immunization, adverse events, communication

## Abstract

Introduction: Human behavior and understanding of the vaccine ecosystem play a critical role in the vaccination decision-making process. The objective of this study was to understand different cognitive biases that may lead to vaccine acceptance or hesitancy. Methods: The eligibility criteria for this scoping review was vaccination-related cognitive bias studies published in the English language from inception to April 2022 and available on PubMed, Embase, and Google Scholar. It included all geographical locations and individuals of all age groups and excluded studies focusing on (i) clinical trials of vaccines, (ii) vaccine research conduct bias, (iii) cognitive delay, or (iv) statistical biases. The search method also included reviewing references in the retrieved articles. Results: Overall, 58 articles were identified, and after screening, 19 were included in this study. Twenty-one cognitive biases with the potential to affect vaccination decision-making were observed. These biases were further grouped into three broad categories: cognitive biases seen while processing vaccine-related information, during vaccination-related decision-making, and due to prior beliefs regarding vaccination. Conclusions: This review identified critical cognitive biases affecting the entire process of vaccination that can influence research and public health efforts both positively and negatively. Recognizing and mitigating these cognitive biases is crucial for maintaining the population’s level of trust in vaccination programs around the world.

## 1. Introduction

Routine childhood vaccination was one of the most cost-effective and life-saving public health interventions in the 20th century. According to the World Health Organization (WHO), vaccination prevents millions of cases of vaccine-preventable disease-related morbidity and averts 2.5 million deaths each year [1]. Consideration of the vaccination decision-making process is crucial for drawing conclusions about the dynamics of vaccination acceptance or rejection in the context of whether voluntary vaccination programs succeed or fail by influencing that decision-making [2]. Over the past decade, several theoretical-based studies have shown that human behavior is one of the main determinants shaping how vaccine concerns manifest [3,4]. This has contributed to the new area of behavioral epidemiology and the development of “prevalence-based” paradigms [4], which postulate that people modify their choices depending on how they receive, perceive, and process information [5]. A number of factors, including prior immunization experiences, faith in the government and medical establishment, parental attitudes, and prevailing socio-cultural norms, among others, contribute to varying perspectives on vaccination, which may be independent of more traditional science- or evidence-based vaccine decision-making considerations like disease epidemiology, risk–benefit, or cost–benefit analysis [2]. One example could be recent parental decisions to not have their children vaccinated with vaccines containing measles, either due to the strong influence of social media or the resurfacing of old misinformation related to safety [6,7].

Numerous cognitive biases and mental heuristics associated with decision-making have been described in prior research. A strong preconceived view about someone or something that is on the basis of information that we either possess, think we possess, or do not possess is referred to as a cognitive bias [8]. In order to expedite information processing and help with the quick interpretation of visual data, the human brain develops certain presumptions as mental shortcuts. A person’s views, observations, or points of view may lead to any number of cognitive biases, which are systemic errors in that individual’s style of thinking. People experience a variety of biases, and these biases have an effect on our thoughts, behavior, and decision-making. People find it challenging to communicate accurately or get to the truth when they are biased. Our ability to think critically is distorted by cognitive biases, which may help spread false information or detrimental misunderstandings [8]. Biases cause people to avoid information that can be upsetting or challenging rather than looking into the facts to help make better-informed decisions. Furthermore, biases may cause people to see connections or correlations between ideas that are not truly there.

Bias is the unjust support of or opposition to a certain person or thing owing to the influence of personal beliefs on decision-making [9,10]. A better understanding of the impact of cognitive bias on vaccine acceptance/reluctance could be valuable in enhancing the uptake of recommended vaccines. Although anecdotal reports of cognitive biases around vaccination have been observed, there has been a dearth of published reviews on the possible impact of cognitive biases on vaccination dynamics.

A thematic analysis of the data in the context of current immunization strategies and a scoping review of the possible cognitive biases impacting vaccination are necessary for the evaluation of the knowledge gap. In this article, we review existing literature with the intent of understanding the theoretical investigations carried out on the effects of cognitive bias on overall vaccination. As part of the review, we provide a practical categorization scheme to assist with organizing the diverse range of cognitive bias concepts. The second goal is to evaluate the impact of cognitive bias on knowledge of vaccine dynamics and vaccination coverage, as well as suggestions for responding to cognitive bias and mitigating it.

## 2. Methodology

### 2.1. Study Process

A literature search on evidence pertaining to cognitive biases related to decision-making in vaccination was conducted. Once all studies had been identified and examined by the investigators, a summary of the evidence was developed. The scoping review approach provided by Arksey and O’Malley, which was further modified by Levac et al., was used for this study [11,12]. This approach allowed us to search from a wider range of articles pertinent to the research question. The scoping review was conducted from 1 April 2022, to 31 December 2022.

### 2.2. Search Strategy

MeSH terms including (“Vaccination” OR “Immunization”) AND (“Cognitive Bias” OR “Biases” OR “Cognitive Distortion”) were used in different combinations using Boolean operators to search articles for this study. The search strategy was designed in consultation with experts. The next step was “snowballing,” which included iteratively searching all full-text article reference lists and current reviews for any new publications that could have been appropriate for inclusion. A rigorous iterative process was used to create this review. The cited references were searched using Google Scholar and Medline, and the “similar articles” section was checked as part of the citation-tracking process.

### 2.3. Databases

Database searches were conducted mainly in three databases (PubMed, Embase, and Google Scholar) using the abovementioned search strategy. Citations from selected articles were also reviewed for possible additional references and a supplementary manual search was conducted on Google (Appendix A).

### 2.4. Study Selection

There was no restriction on the age of the research participants or location (geographical distribution) restrictions for the study selection. The following criteria for inclusion had to be met by potential eligible articles: (i) published in or before April 2022, (ii) full-text articles that were developed and published in English, (iii) published articles including cognitive bias-related factors, and (iv) studies related to vaccination. Articles were excluded if they concentrated on (i) clinical trials of vaccines, (ii) vaccine research conduct bias, (iii) cognitive delay, or (iv) statistical biases (Figure 1).

### 2.5. Data Extraction

The qualitative data were extracted by A.R. and reviewed by A.K.S. After the initial evaluation, any disputes were addressed and settled by mutual discussion or consultation with A.L.W. and M.L.B. This review addresses the factors and biases that were most extensively prevalent.

### 2.6. Analysis and Data Items

Given that this was a scoping review whose primary goal was to outline the existing literature on cognitive biases impacting vaccination, a qualitative narrative synthesis was conducted.

These cognitive biases were categorized into the following three broad categories:

Group #1: Cognitive biases seen during processing vaccine-related information;

Group #2: Cognitive biases seen during vaccination-related decision-making;

Group #3: Cognitive biases due to prior beliefs regarding vaccination.

This categorization is based on the common factors in each group. The cognitive biases in group #1 are those that are heavily dependent on the message, the content of the message, and the relevant factors such as its framing and the emotions that have a significant effect. The cognitive biases in group #2 are those that are triggered by the factors that are most prominent when people are in the decision-making process. When making a vaccine-intake decision, a person’s subjective assessment of their own risk is a significant factor in the process. In this situation, people’s ability to assess the risk is constrained, and they could be unclear about the outcome. People’s decisions are most significantly affected by group #2’s cognitive biases when they encounter uncertainty, ambiguity, risk perception, and other factors. Cognitive biases associated with preexisting beliefs about vaccinations were identified in group #3 [13]. Here, people’s actions are more influenced by their preexisting ideas than by the information they are being provided. Because of the cognitive dissonance caused by the new contradicting information, decision-makers at this point typically stick to their original beliefs [14].

### 2.7. Identification of Knowledge Gaps

Different cognitive biases in vaccination were identified, and potential biases were line-listed. An evaluation of knowledge gaps was then performed. Further development focused on a methodology for prioritizing knowledge areas that need investigation.

## 3. Results

Overall, 24 cognitive biases were found to be have the potential to affect the vaccination process.

### 3.1. Group #1: Cognitive Biases Seen during Processing of Vaccine-Related Information

The *framing effect* is the phenomenon wherein an individual’s decision is influenced by the presentation of communication content without changing its main message, even when the outcomes of two different programs were the same—for example, participants in a study by Tversky and Kahneman preferred the program that was presented in the risk-averse frame to the program that was presented in the risk-taking frame [15,16]. The framing effect, which increases vaccination acceptability by favorably narrating messages, has been highlighted in the literature. The opposite is also true, too, since vaccine-related material and immunization results that include vaccine reluctance are adversely framed by anti-vaccine science people [14].

The tendency to focus on specific information while disregarding general information, even when the general information may be more important, is known as *base rate neglect* [17]. An instance of this would be the overestimation of rare serious/non-serious adverse events following immunizations (AEFIs) and the underestimation of common mild AEFIs. Even though there is a much smaller chance of a rare serious/non-serious AEFI than a common mild AEFI, when evaluating the risk of vaccines, potential vaccine beneficiaries frequently ignore denominators. *Base-rate neglect*, which is caused by individuals having trouble grasping ratio concepts, is the main factor when two sides of an issue are being discussed in the same context. However, additional variables enter the picture and contribute to other cognitive biases when a single tale of a serious AEFI overcomes the majority of mild AEFIs. For instance, despite its low likelihood, a personal account of a single child who has experienced an AEFI is far more effective than a much larger number of mild AEFIs. A phenomenon known as availability bias occurs in these situations, wherein emotions also play a role in giving decision-makers a compelling story to consider.

*Availability bias* is the tendency to assign greater weight to elements that are simpler to recall. The media’s portrayal of an unusual serious AEFI report that conveys a strong and compelling anti-vaccine message and is likely to stick in people’s minds when making decisions may cause people to overestimate the possibility of an AEFI [14].

*First-impression bias*, also referred to as the anchoring effect, is the tendency to make a decision heavily based on a value that is originally presented [18]. Clinicians may be persuaded to recommend the human papillomavirus (HPV) vaccination to patients based on their age or other physical characteristics, such as their pubertal status. Such initial perceptions might hinder people’s subsequent cognitive processing and may encourage greater vaccination hesitancy.

The propensity to give authoritative persons’ opinions greater weight is known as *authority bias.* Depending on the position taken by the relevant authorities, it might be used either in favor of or against vaccination. In this context, medical experts are considered authoritative since they are trustworthy providers of knowledge regarding vaccines. In general, they are pro-vaccination, and their influence works in their favor. However, spreading misinformation against vaccination from a reputable source may also influence people’s decisions and make vaccination acceptance less likely. A tendency towards general distrust in healthcare workers or vaccine products and a focus on negative aspects could be considered negative bias. It could be due to range of factors, including historical events and personal experiences (Table 1).

### 3.2. Group #2: Cognitive Biases Seen during Vaccination Decision-Making

When a person chooses not to take a certain action (omission) rather taking it (commission), even when the consequences of omission are greater than or equal to those of commission, they are exhibiting omission bias, which is the tendency to underestimate the severity of consequences [20,28,29]. Ritov and Baron examined the effect of omission bias on parents’ vaccine hesitancy and found that parents had a high propensity to omit when vaccinations could result in AEFIs [30]. Parents believe that the adverse events of vaccinations are substantially more severe and long-lasting than the medical sequelae of a sickness. Due to factors like expected responsibility and regret, decision-makers have a strong tendency to neglect vaccinations. Additionally, availability bias makes omission bias worse by making the negative effects of a choice more readily apparent to decision-makers. Vaccine-skeptical individuals may have easier access to incomplete information on reports of serious AEFIs, which might lead to availability bias and drive decision-makers to make omissions.

The propensity to choose a known risk over an unknown danger, no matter the results, is known as ambiguity aversion. People who choose a known danger from a disease over a vaccination’s more unclear risk experience ambiguity aversion, which is one possible cause of vaccine hesitancy. *Omission bias* is further exacerbated by ambiguity on its own. When there is greater uncertainty about the outcome of immunization, decision-makers would rather not commit than omit [31]. The propensity to prioritize averting losses above gaining equal rewards is known as loss aversion. Patients may only concentrate on a 1% probability of experiencing AEFIs while discussing AEFIs rather than the 99% possibility of not experiencing an AEFI. The same principle applies when assessing vaccination outcomes in terms of commission (vaccinating) and omission (not vaccinating). The aversion to commission loss is higher than that to omission loss. Optimism bias is the tendency to have an extremely positive outlook on a particular health risk and believe that it will more likely affect other people than oneself [32]. It is equivalent to assuming that one is less likely than others to contract a vaccine-preventable disease (VPD) [33]. In the case of the COVID-19 pandemic, for instance, optimism bias led individuals to believe they were healthy, resistant to the disease, and capable of fighting it off. Another example is how physicians’ confidence in their patients’ minimal risk of contracting HPV stems solely from their established personal connect with patients and their families.

The propensity to prioritize current expenses and advantages above those that will be achieved in the future is known as *present bias.* People place higher importance on vaccine side effects and expenses since they are visible to the public and available to decision-makers. Future benefits, which might not be very common and are therefore given less weightage, include protection against vaccine-preventable disease. People reject making trade-offs against protected ideals because they are absolute and impervious to interference no matter the consequences. Repercussions are any possible drawbacks from not receiving a vaccination, whereas *protected values* are any opinions that vaccination contradicts. The subjects do not compromise such values, regardless of the size of the cost, the magnitude of the gain, or the severity of the repercussions. A few examples of protected values are the ability of parents to object to vaccinations and the fact that males are not required to receive the HPV vaccine. Protected values have also been cited as an explanation for omission bias since they make people more willing to withhold information even when doing so would be detrimental to them. Short pieces of information on the *Dunning*–*Kruger effect* and *status quo* biases are provided in Table 2.

### 3.3. Group #3: Cognitive Biases Due to Prior Beliefs Regarding Vaccination

The strongest influence on vaccine-hesitant individuals is from group #3 of cognitive biases, which also makes them harder to persuade and more likely to stick with their original decision (to not get vaccinated) [14]. *Confirmation bias* is the tendency to only consider information that confirms our preconceived notions. It thwarts efforts to refute incorrect information that vaccine-skeptical individuals have because they often disregard data that contradict their ideas. It leads anti-vaccine individuals to overestimate AEFIs and downplay the threat of a VPD.

The propensity to judge an argument’s validity based on how credible its conclusion is known as *belief bias.* It hinders people’s cognitive capacities when the understanding of new information conflicts with their pre-existing views. Arguments raised by anti-vaxxers cover a broad spectrum, including vaccination efficacy and safety, alternative medicine, conspiracy theories, civil rights, morality, ideology, and religion [42]. *Confirmation bias* discourages individuals from paying attention to newly contradicting information after having been exposed to and believing in alternate facts. Even after people interact with such material, belief bias limits their capacity to critically evaluate new information.

The propensity to focus more time and effort on information that group members already know and less time and effort on new information is known as *shared information bias*. The following conversations and debates revolve around the misinformation about vaccines that people in anti-vaccination organizations disseminate on social media. When confronted with and processing new information, members of such organizations may exhibit biases, including *belief bias* and *confirmation bias.* It also involves a bias caused by shared knowledge, which might make the *false consensus effect* worse. The *false consensus effect* refers to the tendency to overestimate the degree to which one’s opinions are held by the general public (believing that they are more widely held than they actually are) [43]. Mothers who are averse to vaccination are more likely to discuss the topic on social media [44]. Mothers who favor vaccination, however, are less likely to take part in such conversations. Poorly informed mothers that become involved in online discussions foster tiny but potent online anti-vaccination communities that lead to attitudes and behaviors that support anti-vaccination ideas. These networks possess a high degree of false consensus regarding vaccine-hesitancy issues.

Other cognitive biases, such as default and the bandwagon effect, might aid in the explanation of vaccination reluctance, depending on how they are interpreted, in addition to the cognitive bias that may lead to vaccine hesitancy. A propensity towards the default option when presented with multiple options is known as the *default effect* [45]. More individuals prefer to select vaccination when it is made the default option. The propensity to follow the majority of other individuals in their decisions is known as the “bandwagon effect.” When something is defined as “jumping on the bandwagon,” it can be because individuals believe others have already made a sensible choice or because of peer pressure. Depending on how the information is presented, it may be a key motivation for choosing a vaccine. For instance, emphasizing the societal pressure to immunize makes the bandwagon effect clear. However, the bandwagon loses its effectiveness if the material places more emphasis on other topics, such as community immunity [14]. Details of cognitive dissonance, illusory correlation, and cognitive biases that appeal to nature are presented in Table 3.

## 4. Discussion

Public health professionals may use the categorized information and the observed cognitive biases to promote vaccine acceptance and confidence. Public health authorities may tailor their strategies, interventions, and other kinds of communication based on the categories presented in this review to downgrade the effects of cognitive biases associated with poor vaccination decisions. In group #2, cognitive biases may be considered first to be targeted. An attempt may be made to improve vaccine decision-makers’ perceptions of uncertainty, ambiguity, and loss in relation to the outcomes of immunization. Program managers may utilize the specific tools to address the cognitive biases in group #2 as well as mechanisms to inform caregivers about vaccine safety and adverse events (AEs) based on the existing evidence in a manner that enhances confidence in immunization programs.

Both immunization program champions and people who think negatively about vaccines use the cognitive bias in group #1. Therefore, communication campaigns may be motivated to address the cognitive biases in group #1, which are common in anti-vaccine content. During the course of the COVID-19 pandemic there was an infodemic on social media. Health authorities were at the forefront of managing this infodemic. However, as it was an evolving situation, there were certain challenges like timely response to any miscommunication. Furthermore, evolving understanding of the disease transmission mode, the concept of herd immunity, reinfection, and breakthrough infection created doubts in people’s minds. Initiatives to increase the accessibility information about the benefits of vaccines and disseminate information on more prevalent AEFIs could help lessen the effects of base-rate neglect. More crucially, efforts may be considered to increase trust in vaccines by using biases like the framing effect and authority bias. For women with low involvement, such as those who do not currently have children or plan to have children, it is more advantageous to frame adverse effects of vaccination positively, as a high likelihood of no side effects. In the United States, the Centers for Disease Control and Prevention’s successful efforts to combat misinformation about health on social media may be an example of authority bias [46,56]. Careful consideration must be given to reducing the impact of group #3’s cognitive biases, which are held by vaccine-hesitant individuals. One method that is generally accepted is to avoid outright rejection or debunking of false information, as that might have the reverse effect since direct refutation necessitates reiterating falsehoods. Repeating something makes false information easier to remember, which has a beneficial impact on confirmation bias.

Although it is not possible to completely counter preconceived notions and beliefs, there are techniques to minimize them. Raising awareness among the public may lead to people identifying their own biases. Simply accepting that none of us can avoid biases would make it easier to include extra measures and procedures to address them. Furthermore, an Institute for Government study about policy talks listed three points on the influence of group opinion that are valuable for policymakers and stakeholders: (a) Early contributions have a powerful influence on group consensus, (b) there is a tendency for groups to concentrate on what most members already know, and (c) there is potential for discussion to amplify the extreme viewpoints of a group [57]. People also have the propensity to solve problems even before they have been accurately identified and mentally processed. Given this, more organized sessions may provide a fairer and more equitable forum where the most outspoken or influential individuals do not dominate discussions [58]. However, ideally, “critical friends” may offer challenging questions, uncover latent assumptions, challenge collective thinking, and balance prevailing voices. Having a variety of individuals with various opinions does not always guarantee effective countering of pre-conceived biases. Individuals or members of a group may also serve as “red teams” or “devil’s advocates.” They may provide counterarguments and challenge our reasoning when given a clear directive, but they could otherwise be seen as unfriendly or unconstructive. This might have a particularly major impact since individuals are more inclined to accept criticism coming from someone in their own social circle [59]. However, it is more crucial than ever to be aware of these prejudices given worries about the polarization of public discourse and how the internet magnifies such biases [60]. There is a need to establish processes, programs, and institutions that encourage critical thinking and consideration of a variety of viewpoints before, during, and after sessions of collaborative analysis and decision-making. By doing this, some of these biases may be reduced or even eliminated, and analysis and judgments can make coverage of vaccine acceptance more likely.

## 5. Limitations

Despite our efforts to detect cognitive biases related to vaccination and decision-making about vaccination in accordance with the guidelines of a scoping review, the outcome cannot be regarded as a systematic review. The proposed remedies for cognitive biases, the underlying causes of each cognitive bias and how they affect behavior, and the specific mechanisms by which the cognitive biases create transitions between phases have not been discussed. This may also apply to elements not included in our research, such as proposed therapies, the root causes of cognitive biases, transitional processes between phases, and the use of other grounded health-behavior models. Furthermore, the articles that were subsequently cited and included in this review may have been influenced by publication bias. Also, this study did not include any quality assessment of biases in the included articles.

## 6. Conclusions

Vaccine reluctance has become an increasingly pressing public health challenge that has been observed during the course of the COVID-19 pandemic and continues to pose a challenge. In order to promote innovative approaches to increasing vaccine acceptability and building confidence in the community, an understanding of the impact of relevant cognitive biases is important. This review has attempted to highlight possible cognitive biases that influence vaccination decision-making and communication. These findings would be beneficial for immunization program managers, policymakers, and immunization communication officials to devise multipronged communication strategies to address cognitive biases associated with an intent to boost vaccine acceptance and improve confidence in vaccine uptake decision-making by parents, the elderly, and adolescents. The cognitive biases identified to be part of three broad categories (seen during the processing of vaccine-related information, seen during vaccination-related decision-making, and due to prior beliefs regarding vaccination) could be helpful in exploring their relative influence on immunization acceptance or aversion in future studies.

## Figures and Tables

**Figure 1 vaccines-11-01837-f001:**
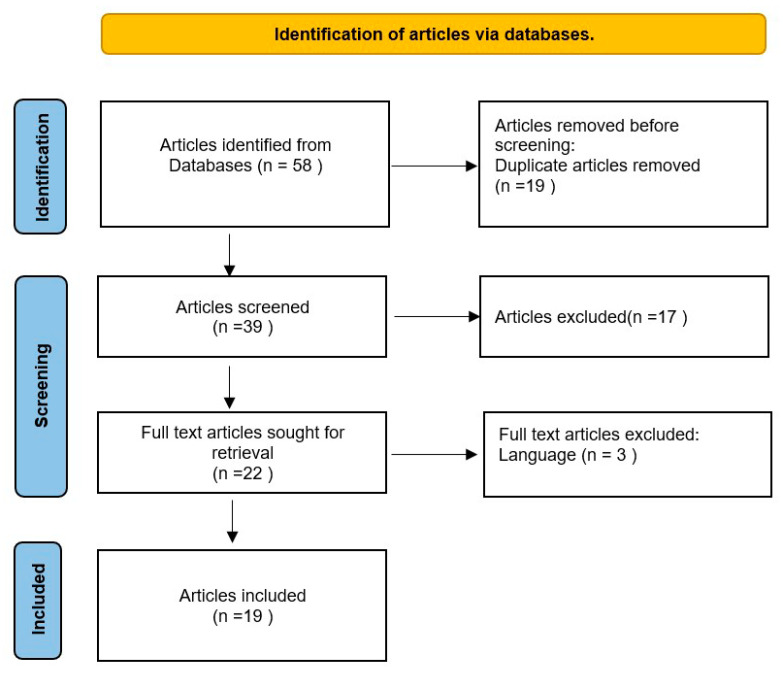
Flowchart of the scoping review of cognitive biases related to immunization.

**Table 1 vaccines-11-01837-t001:** Cognitive biases seen during the processing of vaccine-related information.

No.	Cognitive Bias	Definition	Example
1	Framing effect	The agent’s decision is influenced if you compose a message without changing the primary message [15].	By highlighting the lesser proportion of patients with AEFIs than the majority of patients with no AEFIs, one might cast doubt on the effectiveness of vaccination.
2	Base-rate neglect	The inclination to prioritize specialized information while ignoring broad information despite the fact that the latter is more crucial [19].	Rare AEFIs are overestimated, whereas typical, moderate AEFIs are underestimated [20].
3	Availability bias	The inclination to give elements that are simpler to remember more weight [21].	The media coverage of a rare serious or severe AEFI incident provides a dramatic and emotionally stirring message that is likely to be remembered when vaccination decisions are made [22].
4	Anchoring effect	The capacity for making decisions that largely depend on values that are originally offered [23].	One sees a side effect after vaccination and thinks that vaccines with that particular side effect are more prevalent [24].
5	Authority bias	The tendency to give the opinions of people of authority greater weight [25].	When a medical practitioner disseminates anti-vaccination material, it may influence individuals to choose not to be vaccinated since the medical practitioner is an authoritative person.
6	Pessimism bias	The propensity to overestimate the chance of bad things happening while underestimating the likelihood of good things happening is known as pessimistic bias [26].	Children will likely not have an AEFI after receiving a vaccine, but anxious/panic/depressed parents may think they will.
7	Negativity bias	More trust is given to negative information than positive information [27].	More focus is placed on rare adverse events associated with vaccines than their overwhelming benefits.

**Table 2 vaccines-11-01837-t002:** Cognitive biases seen in vaccination decision-making.

No.	Cognitive Bias	Definition	Example
1	Omission bias	The propensity to undervalue the consequences of taking action (commission) even when the consequences of inaction are worse or on equal to those of action (omission) [29].	When parents foresee AEFIs, they prefer to omit immunization because they view it as a commission (not vaccinating).
2	Ambiguity aversion	The propensity, regardless of consequences, to choose a known danger over an unknown risk [34].	People choose established risks from diseases over more uncertain risks associated with vaccination against the same disease [35].
3	Loss aversion	The propensity to place more importance on preventing losses than making equal gains [36].	Patients may only concentrate on a 1% probability of experiencing AEs while discussing AEFIs rather than the 99% possibility of no AEs [37].
4	Optimism bias	The propensity to view a specific health issue with an overly positive outlook and believe that others face it more seriously than oneself [22].	People believe they are healthy, immune to the flu, and able to fight it off, so they do not think of themselves as being at danger of contracting it [38].
5	Present bias	The tendency to prioritize current expenditures and advantages above those obtained in the future [22].	People are more aware of the adverse reactions to vaccines (as a cost); thus, they are given greater weight. Future benefits that are not immediately obvious are given less weight, such as immunity to a disease.
6	Protected values	Protecting absolute ideals that individuals believe should not be sold off should not be a priority [30].	Respecting parents’ choice about vaccination [13].
7	The Dunning–Kruger effect	A cognitive bias in which individuals with relatively poor intellectual or social ability substantially overestimate their own knowledge or competence in that subject in comparison to external standards, the performance of their peers, or that of the general population [39].	Anti-vaccination policy attitudes [40].
8	Status quo bias	When someone prefers to do nothing or adhere to a past choice, it is clear that they are biased [41].	Unvaccinated children remain unvaccinated due to parents’ status quo thinking.

**Table 3 vaccines-11-01837-t003:** Cognitive biases due to prior beliefs regarding vaccination.

No.	Cognitive Bias	Definition	Example
1	Confirmation bias	The propensity to remember and understand data that support our preexisting ideas [46].	People who are vaccine-hesitant exaggerate AEFIs and downplay the threat of diseases that may be prevented by vaccination [5,47]. People tend to focus on what matters to them and ignore what does not, which often results in the “ostrich effect,” in which a person buries their head in the sand to avoid facts that would contradict their initial assertion.
2	Belief bias	The propensity to assess the validity of an argument is dependent on the conclusion’s plausibility [48].	It would be ineffective to discuss vaccine safety in terms of minor AEFIs with those who think vaccination programs are driven by huge businesses’ profits.
3	Shared information bias	The propensity to focus more time and effort on material that group members are already acquainted with while spending less time and effort on fresh information [49].	Concentrating on just a few anti-vaccine issues, such as the disproved MMR–autism connection on internet anti-vaccine echo chambers.
4	False consensus effect	The propensity to exaggerate how much one’s viewpoint is shared by the wider public [50].	Mothers who are against (for) vaccination are more (less) likely to discuss the topic on social media [51]. This leads to the development of strong false consensuses on vaccine reluctance in online groups.
5	Cognitive dissonance	Cognitive dissonances are beliefs, attitudes, or behavior that clash with each other [52]. One of these attitudes, beliefs, or behaviors changes as a reaction to the mental discomfort that results from this in an attempt to reduce the discomfort and restore balance.	If a parent learns that vaccinations are effective but is also concerned that they can endanger their kid, they may conclude that vaccines do not function in order to get rid of the cognitive dissonance.
6	Illusory correlation	The illusory correlation is the assumption of a link between two variables when it is likely not true [53].	Any instance of autism and vaccination co-occurring is used by parents to justify their decision to not vaccinate because they have a preconceived notion that there is a link between vaccination and autism.
7	Appeal to nature bias/fallacy	When it is suggested that something is good because it is natural or bad because it is unnatural, there is bias involved [54]	Some individuals place a higher value on innate immunity than artificially induced immunity boosters like vaccinations [55]

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
