# Peer review of "Mapping the Cognitive Biases Related to Vaccination: A Scoping Review of the Literature"

_vaccines, 2023, doi:10.3390/vaccines11121837_

Round 1
Reviewer 1 Report
Comments and Suggestions for Authors
GENERAL
- - There was no data regarding the registry of this work
- The author did not mention PRISMA as reporting guidelines for the review. There was PRISMA in the reference list, but PRISMA has a specific form for scoping review, and the author did not mention this form (The original publication of PRISMA for scoping review was made by Triccio, et al.)
ABSTRACT
- Methods: Exclusion criteria may be mentioned
- Methods: What kind of extracted data was collected?
- Results: I suggest the author mention the three kinds of biases
- Keywords: Was it necessary to mention both immunization and vaccination?
INTRODUCTION
- Lines 42-44 needed references
METHODS
- Was there any quality assessment of the retracted articles?
- Line 102-106: What kind of extracted data in this study
RESULTS
- I suggest the author make a table which consists of the list of the articles with the specific characteristics.
- Was there any bias in those selected manuscripts?
CONCLUSION
- Three biases may be written in the conclusion
REFERENCES
- Few typos, such as numbers 10 and 21
FIGURES
Figure 1: 17 articles were excluded because of what?
Comments on the Quality of English LanguageI do not have any comment on the language
Author Response
Thank you very much for taking the time to review our manuscript with following title: Mapping the Cognitive Biases related to Vaccination: A scoping review of the literature. Please find the detailed responses below and the corresponding revisions/corrections highlighted/in track changes in the re-submitted files.
Comment 1: The author did not mention PRISMA as reporting guidelines for the review. There was PRISMA in the reference list, but PRISMA has a specific form for scoping review, and the author did not mention this form (The original publication of PRISMA for scoping review was made by Triccio, et al.)
Response 1: Many thanks for pointing out. This study didn’t not include PRISMA guidelines for the scoping review because that restricts flexibility to a checklist. This study utilized scoping studies guidelines by Arksey and O'Malley that were modified by Levac et al. That give more flexibility given the nature of this scoping review. However as per the advice PRISMA checklist has been filled.
PRISMA was in reference list due to flowchart. In order to avoid confusion it has been removed from reference.
Abstract
Comment 2.1: Methods: Exclusion criteria may be mentioned
Response 2.2: Updated at line # 15-16 in the revised manuscript
Comment 2.3:Methods: What kind of extracted data was collected?
Response 2.3: Qualitative content was extracted
Comment 2.4: Results: I suggest the author mention the three kinds of biases
Response 2.4: Updated at line # 20-22 of the revised manuscript
Comment 2.5: Keywords: Was it necessary to mention both immunization and vaccination?
Response 2.5: As both of them carry different meaning it is worth to keep both of them.
INTRODUCTION
Comment 3.1: Lines 42-44 needed references
Response 3.1: Added at line # 51 of the revised manuscript
Methods
Comment 4.1: Was there any quality assessment of the retracted articles?
Response 4.1: As primary focus of the scoping review was to provide broad overview of existing literature and mapping key cognitive biases related to vaccination. A quality assessment of retracted articles was out of scope of this study.
Comment 4.2: Line 102-106: What kind of extracted data in this study
Response 4.2: Qualitative content relevant to the objective of the study was extracted.
RESULTS
Comment 5.1: I suggest the author make a table which consists of the list of the articles with the specific characteristics.
Response 5.1: Objective of the study is to map out different cognitive bias that we have summarized in the table with references
Comment 5.2: Was there any bias in those selected manuscripts?
Response 5.2: As stated earlier, objective was to present overview pf available literature, Individual article quality assessment was out of scope of this study
CONCLUSION
Comment 6: Three biases may be written in the conclusion
Response 6: Updated at line # 387-389 of the revised manuscript
REFERENCES
Comment 7: Few typos, such as numbers 10 and 21
Response 7: Addressed the formatting error
FIGURES
Comment 8: Figure 1: 17 articles were excluded because of what?
Response 8: Based on the exclusion criteria mentioned in the methodology. A thorough review of well-established and commonly agreed upon cognitive biases as characterized in the peer review literature guided our decision regarding which to include and which to exclude.
Reviewer 2 Report
Comments and Suggestions for Authors
General feedback
This study reported a scoping review conducted from April 01, 2022, to December 31, 2022, to examine cognitive biases (categorized into 3 main groups), that may lead to vaccine hesitancy or acceptance.
The manuscript is well written and addresses important points in the attitude of people towards vaccination.
Specific points
· The second paragraph of introduction needs citations
· Scopus not considered as searching engine?
· Line 92-93: “and a supplementary manual search on Google search was conducted”, “search” is redundant and could be removed from “Google search”
· When discussing authoritative bias, the authors may want to include also lack of trust on health care workers by conspiracysts, who may view health care workers as representatives of the industry or governments.
· In discussion, when discussing group #1 cognitive bias and communication campaigns, the authors may also want to add that in in order to enforce vaccination during the pandemic, governments used the improper argument of “heard immunity” against SARS-CoV-2, for which there was still no evidence when vaccines were authorized, as the respective phase 3 clinical trials only measured prevention of symptomatic disease, not (asymptomatic) infection. This contributed to spread conspiracy theories and mistrust towards government officials and health care professionals. Although, subsequent observational studies reported vaccine effectiveness also to prevent asymptomatic infections and reduce viral shedding time among vaccinated [useful citations: PMID: 37515237; PMID: 37242504, DOI: 10.3390/v15112180].
· Lines 304: “more conscious of biases among people”, maybe “awareness of biases”
Author Response
Thank you very much for taking the time to review this manuscript. Please find the detailed responses below and the corresponding revisions/corrections highlighted/in track changes in the re-submitted files.
Comment 1: The second paragraph of introduction needs citations
Response 1: As suggested, citation has been included
Comment 2: Scopus not considered as searching engine?
Response 2: It was not accessible during the period of the study
Comment 3: Line 92-93: “and a supplementary manual search on Google search was conducted”, “search” is redundant and could be removed from “Google search”
Response 3: Deleted
Comment 4: When discussing authoritative bias, the authors may want to include also lack of trust on health care workers by conspiracysts, who may view health care workers as representatives of the industry or governments.
Response 4: It may not come under authority bias because that is related to tendency of individuals to attribute greater accuracy, expertise, or trustworthiness to the opinions, advice, or directives of an authority figure. This issue of distrust in authority could be more related to negative bias. Where people tend to have general tendency of negative information. It could be due to multiple factors, like an historical event or personal experience. Added under negative bias on line #205-208
Comment 5: In discussion, when discussing group #1 cognitive bias and communication campaigns, the authors may also want to add that in in order to enforce vaccination during the pandemic, governments used the improper argument of “heard immunity” against SARS-CoV-2, for which there was still no evidence when vaccines were authorized, as the respective phase 3 clinical trials only measured prevention of symptomatic disease, not (asymptomatic) infection. This contributed to spread conspiracy theories and mistrust towards government officials and health care professionals. Although, subsequent observational studies reported vaccine effectiveness also to prevent asymptomatic infections and reduce viral shedding time among vaccinated[useful citations: PMID: 37515237; PMID: 37242504, DOI: 10.3390/v15112180].
Response 5: Added information at line 318-323 of the revised manuscript
Comment 6: Lines 304: “more conscious of biases among people”, maybe “awareness of biases”
Response: As advised it has been rephrased.
Reviewer 3 Report
Comments and Suggestions for Authors
After reading this manuscript. The content in this review covers the aims and scope of the bias related to vaccination. The literature on search strategy is well-conducted and well-summarised.
Comments.
1. An example of misinformation driven by the outbreak in any part of the world is "measles"
Recently, measles outbreaks in many parts of the world, including Europe, Japan and the USA. These countries have vaccines accessible to all people.
The vaccine hesitancy could drive parents not to give a vaccination to their child.
I think you can add it to the introduction or discussion.
Example literature;
Ashkenazi S, Livni G, Klein A, Kremer N, Havlin A, Berkowitz O. The relationship between parental source of information and knowledge about measles / measles vaccine and vaccine hesitancy. Vaccine. 2020 Oct 27;38(46):7292-7298. doi: 10.1016/j.vaccine.2020.09.044. Epub 2020 Sep 25. PMID: 32981777.
Novilla MLB, Goates MC, Redelfs AH, Quenzer M, Novilla LKB, Leffler T, Holt CA, Doria RB, Dang MT, Hewitt M, Lind E, Prickett E, Aldridge K. Why Parents Say No to Having Their Children Vaccinated against Measles: A Systematic Review of the Social Determinants of Parental Perceptions on MMR Vaccine Hesitancy. Vaccines (Basel). 2023 May 2;11(5):926. doi: 10.3390/vaccines11050926. PMID: 37243030; PMCID: PMC10224336.
Author Response
Comment 1: An example of misinformation driven by the outbreak in any part of the world is "measles" Recently, measles outbreaks in many parts of the world, including Europe, Japan and the USA. These countries have vaccines accessible to all people. The vaccine hesitancy could drive parents not to give a vaccination to their child. I think you can add it to the introduction or discussion.
Example literature;
Ashkenazi S, Livni G, Klein A, Kremer N, Havlin A, Berkowitz O. The relationship between parental source of information and knowledge about measles / measles vaccine and vaccine hesitancy. Vaccine. 2020 Oct 27;38(46):7292-7298. doi: 10.1016/j.vaccine.2020.09.044. Epub 2020 Sep 25. PMID: 32981777.
Novilla MLB, Goates MC, Redelfs AH, Quenzer M, Novilla LKB, Leffler T, Holt CA, Doria RB, Dang MT, Hewitt M, Lind E, Prickett E, Aldridge K. Why Parents Say No to Having Their Children Vaccinated against Measles: A Systematic Review of the Social Determinants of Parental Perceptions on MMR Vaccine Hesitancy. Vaccines (Basel). 2023 May 2;11(5):926. doi: 10.3390/vaccines11050926. PMID: 37243030; PMCID: PMC10224336.
Response 1: Added in introduction section line #44 -47 in the revised manuscript.